# Ectopic Prostate Tissue in the Uterine Cervix of a Female with Non-Classic Congenital Adrenal Hyperplasia—A Case Report

**DOI:** 10.3390/jcm11154307

**Published:** 2022-07-25

**Authors:** Lea Tschaidse, Matthias K. Auer, Ilja Dubinski, Christian Lottspeich, Hanna Nowotny, Heinrich Schmidt, Nadezda Gut, Nicole Reisch

**Affiliations:** 1Medizinische Klinik und Poliklinik IV, Klinikum der Universität München, LMU München, 80336 Munich, Germany; lea.tschaidse@med.uni-muenchen.de (L.T.); matthias.auer@med.uni-muenchen.de (M.K.A.); christian.lottspeich@med.uni-muenchen.de (C.L.); hanna.nowotny@med.uni-muenchen.de (H.N.); 2Department of Pediatric Endocrinology, Dr. von Haunersches Children’s Hospital, Klinikum der Universität München, LMU München, 80336 Munich, Germany; ilja.dubinski@med.uni-muenchen.de (I.D.); heinrich.schmidt@med.uni-muenchen.de (H.S.); 3Zentrum für Pathologie Allgäu (ZfPA), Medizinisches Versorgungzentrum am Klinikum Kempten, 87439 Kempten, Germany; dr.n.gut@googlemail.com

**Keywords:** 21-hydroxylase deficiency, androgen excess, virilization, ectopic prostate tissue, hyperandrogenemia

## Abstract

Introduction: The occurrence of ectopic prostate tissue in the female genital tract is rare and has only been described sporadically. The origin of these lesions is unclear, but their appearance seems to be associated with various forms of androgen excess, including androgen therapy for transgender treatment or disorders of sex development, such as classic congenital adrenal hyperplasia (CAH). This is the first described case of ectopic prostate tissue in the cervix uteri of a 46,XX patient with a confirmed diagnosis of non-classic CAH due to 21-OHD and a history of mild adrenal androgen excess. Case presentation: We describe a 34-year-old patient with a genetic diagnosis of non-classic CAH due to 21-hydroxylase deficiency (21-OHD) with a female karyo- and phenotype and a history of mild adrenal androgen excess. Due to dysplasia in the cervical smear, conization had to be performed, revealing ectopic prostate tissue in the cervix uteri of the patient. Conclusions: An association between androgen excess and the occurrence of prostate tissue is likely and should therefore be considered as a differential diagnosis for atypical tissue in the female genital tract.

## 1. Background

There have been several reports of prostate tissue or even whole prostate glands in the female genital tract [1,2,3,4,5]. The cause of the development of this tissue is still unclear, since prostatic tissue in women without a history of hormonal, gonadal or genetic disorders has been reported as a very rare incidental finding [2]. Nevertheless, its appearance seems to be influenced by increased androgen concentrations, as several reports show the occurrence of prostatic tissue in the vaginal specimens of patients with different disorders of sexual development (DSD) associated with androgen excess [1,3,4,5] and in 46,XX patients who underwent testosterone therapy due to gender dysphoria [1].

Congenital adrenal hyperplasia due to 21-hydroxylase deficiency (CAH 21-OHD) is an autosomal recessive disorder and one of the most common forms of DSD [6]. It is characterized by cortisol deficiency and an adrenocorticotropic hormone (ACTH)-driven adrenal androgen excess [7]. Based on the clinical manifestation, CAH can be divided into classic, with or without salt wasting, or the non-classic form [8]. With a frequency of about 1:15,000 [8,9], classic CAH is the rarer but much more severe form and is usually diagnosed at birth or during infancy. Due to severe cortisol and possible aldosterone deficiency, the classic form of CAH is a life-threatening disease, making lifelong therapy necessary. Additionally, androgen excess during embryonic development leads to prenatal virilization of the external genitalia in 46,XX patients with classic CAH. The development of internal genital organs is unaffected, as this is determined by genetic sex [7].

The non-classic form, on the other hand, represents the mild variant of the disease and is much more common than classic CAH, with a frequency of 1:200 to 1:1000 [9,10]. Affected patients have sufficient cortisol synthesis and typically develop symptoms of mild androgen excess around the age of puberty, such as acne, hirsutism, alopecia, menstrual irregularities or an unfulfilled desire to have children. The development of external genitalia is normal [11].

In this report, we describe the first case of a patient with a confirmed diagnosis of non-classic CAH due to 21-OHD with a female karyo- and phenotype and ectopic prostate tissue in the cervix uteri. A timeline of the patient’s medical history and care is described in Table 1. Written informed consent was obtained from the patient before submission and publication of this case report. This work was conducted in accordance with the Declaration of Helsinki, and the protocol was approved by the Ethics Committee of the Medical Faculty of Ludwig Maximilians University Munich (Project number 19-558).

## 2. Patient Information

We report on a 34-year-old patient who has been under irregular care of our pediatric and adult endocrinology departments for years due to non-classic CAH. The diagnosis of non-classic CAH was made due to premature development and biochemical presentation. The patient presented to our pediatric clinic for the first time at the age of 10, with an advanced bone age of 14 years, with pubes and breast development on both sides documented as Tanner Stage V and with menarche at the age of 10. Biochemically, the patient showed elevated 17-hydroxyprogesterone (17-OHP) concentrations and hyperandrogenemia with elevated androstenedione (A4), dehydroepiandrosterone-sulfate (DHEA-S) and testosterone concentrations (Table 1; Figure 1). The diagnosis of non-classic CAH was confirmed by genetic testing, showing a compound heterozygous mutation with I2G and P30L mutations. This genotype is traditionally associated with non-classic CAH, but phenotypes may vary [12]. Adrenal insufficiency was excluded. Although the patient had received hydrocortisone therapy with 20 mg hydrocortisone per day between the ages of 10 and 14 years to suppress adrenal androgen synthesis, she continued to demonstrate inadequate hormonal control, as shown in Table 1 and Figure 1. At the patient’s request, hydrocortisone therapy ended after body growth was completed, with a final height of 160.4 cm at the age of 14 years. The patient gave birth to two healthy children at the ages of 27 and 31 years after receiving infertility treatment with clomiphene and under a temporary intake of prednisolone, respectively. After conception of the second pregnancy, the intake of prednisolone was terminated by the patient.

Graphical representation of increased 17-OHP and androgen levels of the patient at different ages, depicted as black data points in relation to sex- and age-specific reference range, shown as a gray area. 17-OHP 17-Hydroxyprogesterone A4 Androstenedione DHEA-S Dehydroepiandrosterone sulfate.

## 3. Clinical Findings

On occasional visits to our clinic, the patient continued to show clinical signs of hyperandrogenemia, such as scarring acne of the upper body and both upper arms, mild hirsutism of the face and mild clitoral hypertrophy. Still, the patient did not wish to receive medication on a regular basis.

## 4. Diagnostic Assessment

At each visit, our patient showed biochemical hyperandrogenemia, documented as the laboratory findings of an untreated, non-classical CAH, with elevated 17-OHP, A4, DHEA-S and testosterone concentrations.

Cervical smears were taken regularly as part of the gynecological check-ups. Recently, a cervical smear revealed an abnormal result with human papillomavirus (HPV)-positive dysplasia. Therefore, conization had to be performed.

## 5. Therapeutic Intervention and Follow-Up

The examination of the portioconisate confirmed a high-grade squamous intraepithelial lesion, as already found in the previously performed cervical smear. However, atypical tissue in the cervix was detected in one quadrant of the conisate. Hematoxylin and eosin (HE) staining showed a normal transition from ecto- to endocervix in most of the specimens, with unremarkable squamous epithelium as well as mucinous epithelium on the surface and in the glands, as depicted in Figure 2A. The atypical metaplastic tissue appeared to be of glandular origin, as depicted in Figure 2A in the HE staining. The atypical glandular proliferations in the endocervix showed the typical morphology of the prostate glands, not the glands of the endocervix. Metaplastic prostate tissue was located outside the endocervical glandular field. The glands showed diffuse positivity for prostate-specific antigen (PSA) and NKX3.1, as depicted in Figure 2C,D. Further immunohistochemical examination of the area showed no expression of p16 and the proliferation rate with Ki 67 was not found to be increased. The morphology of the tissue section and findings on immunohistochemistry support the presence of ectopic, metaplastic prostatic tissue in the cervix uteri, which was resected in total. A subsequent analysis showed a PSA level of 0.013 ng/dL in our patient before conization, which dropped to 0.007 ng/dL afterwards.

## 6. Discussion

The cell morphology, as well as the positivity for PSA and NKX3.1 of the atypical tissue sections found in the patient, is consistent with previous publications on prostate tissue [1,2].

Most reported cases of ectopic prostatic tissue in females are related to a history of severe androgen excess. In particular, several reports of prostate tissue in patients with DSD have been published. Anderson et al. [1] found prostatic tissue in vaginal specimens of multiple patients, one of which likely presented with partial androgen insensitivity due to 46,XY DSD, another with ovotesticular disorder of differentiation and of patients with classic congenital adrenal hyperplasia (46,XX). In 46,XX patients with classic CAH and a male phenotype due to delayed diagnosis, fully developed prostate glands have also been reported [4,5,13,14,15]. Wesselius et al. [3] even described a case of prostate cancer in such a patient, although to our knowledge, this is the only reported case of this kind. However, our patient had only a mild, non-classic form of CAH.

Single cases of prostatic tissue in the female genital tract have even been described in very few patients without any clinical record of hormonal, gonadal or genetic alterations affecting androgen synthesis or effect. McCluggage et al. [2] found ectopic prostate tissue in the lower female genital tract of women who had undergone loop excision due to an abnormal cervical smear. They [2] and others [16] also found ectopic prostate tissue in the cervix uteri. The origin of this prostate tissue is still unknown, but theories of metaplasia and developmental abnormalities have been discussed [16]. Prostate tissue develops from the periurethral glands that occur during early embryonic development in both females and males. Sustained androgen stimulation in males causes these glands to proliferate, whereas in females, they regress due to a lack of stimulus [16,17]. In line, a common feature of the cases described in the literature is elevated androgen levels over a long period of time, either due to increased endogenous androgens [1,4,5] or due to testosterone therapy, e.g., for transgender treatment [1]. Furthermore, the induction of prostatic metaplasia in vaginal tissue through androgens in animal models has been shown [18]. This hypothesis is also supported by the reports of Anderson et al. [1] and Singh et al. [19], in which prostate tissue was found in the lower genital tract of 46,XX patients after having received high-dose testosterone therapy due to gender dysphoria. It has even been suggested that prolonged androgen exposure results in more mature, further differentiated prostate cells [1]. As our patient refused treatment for most of her life, mildly elevated androgen concentrations were likely potential influencing factors in this case. In addition, hyperandrogenaemia continued to be evident during the short time she was receiving drug therapy (Table 1). The location of the tissue in our patient, namely at the cervix uteri, would fit the theory of developmental anomaly. Since the uterus and the urogenital sinus are in close proximity during embryonic development, there could be scattering of the periurethral glands, which could later present as prostatic tissue [16].

Since PSA can sometimes be detected in 46,XX patients with classic CAH [5,20,21] and elevated PSA levels have been described in 46,XX patients with classic CAH and prostate cancer [3,17], we subsequently determined the PSA level from samples before conization and thus removal of prostate tissue in our patient. The PSA level was below the male reference value (<0.03 ng/mL), most likely due to the small amount of ectopic tissue. Nevertheless, PSA should not be underestimated as a biomarker since Paulino Mda et al. [20] found good specificity and sensitivity of PSA for prostatic tissue in girls with classic CAH.

Thus far, prostate tissue has almost exclusively been reported in patients with classic CAH, who—in periods of poor disease control—are exposed to substantially higher androgen concentrations [3,4,5,13,14,15,17,20,21,22,23,24], but not in non-classic CAH. In a study by Paulino Mda et al. [20] prostatic tissue could be detected in five out of 32 girls with classic CAH through MRI, which indicates a prevalence of 15.6% in classic CAH. The number of unknown cases may therefore be significantly higher, as small scattered tissue sections could not be detected through this method. Anderson et al. [1] also described prostate tissue only in the urogenital tract of patients with classic CAH but not of non-classic CAH. The discrepancy between the reported frequency in classic and non-classic CAH could be due to the actual increased occurrence of this tissue in patients with classic CAH or due to selection bias. Since female patients with non-classic CAH do not present with prenatal virilization of the external genitalia, there is no need for genital reconstruction, making surgically removed tissue from these patients for examination hardly available.

To our knowledge, there is only one case report of a 46,XX patient with alleged non-classic CAH and prostate tissue found in the vaginal tissue and cervix uteri by Kim, Park [25]. The female patient presented in young adulthood with an expressed desire for gender reassignment due to a male gender identity. She showed significant clinical virilisation, such as hirsutism, steroid acne, primary amenorrhea and an elongated clitoris of 5 cm. A diagnosis of non-classic CAH was made based on the clinical presentation and one-time elevated laboratory values of ACTH, adrenal androgens and 17-OHP. There was no confirmation of the diagnosis by a short synacthen test or genetic testing [25]. However, given that the patient showed substantial virilization of the external genitalia and highly elevated basal 17-OHP (130 ng/mL), the diagnosis of non-classic CAH is unlikely, but rather classic simple virilizing CAH with presumable higher concentrations of adrenal androgens was the case. Interestingly, other developmental remnants, such as ovarian and testicular adrenal rests, are common in poorly controlled patients with classic CAH due to elevated concentrations of adrenocorticotropic hormones. However, in non-classic CAH, most likely due to only mild hormonal changes, these remnants have not been described.

Thus, to our knowledge, this is the first described case of ectopic prostate tissue in the cervix uteri of a 46,XX patient with a confirmed diagnosis of non-classic CAH due to 21-OHD and a history of mild adrenal androgen excess. As ectopic prostatic tissue could be found in the non-classic form of CAH with only mild androgen excess, we can only hypothesize that the true incidence of prostatic tissue in the female genital tract is likely underestimated. Therefore, the possibility of prostate tissue should always be considered as a differential diagnosis for atypical tissue in the female genital tract, especially in women with a history of some form of androgen excess. In this context, it would be interesting to determine whether ectopic prostatic tissue can also be found in patients with polycystic ovary syndrome. In addition, these case reports emphasize the need for adapted screening in female patients with any form of hyperandrogenemia, as malignant neoplasms of prostate tissue in these patients are possible.

## Figures and Tables

**Figure 1 jcm-11-04307-f001:**
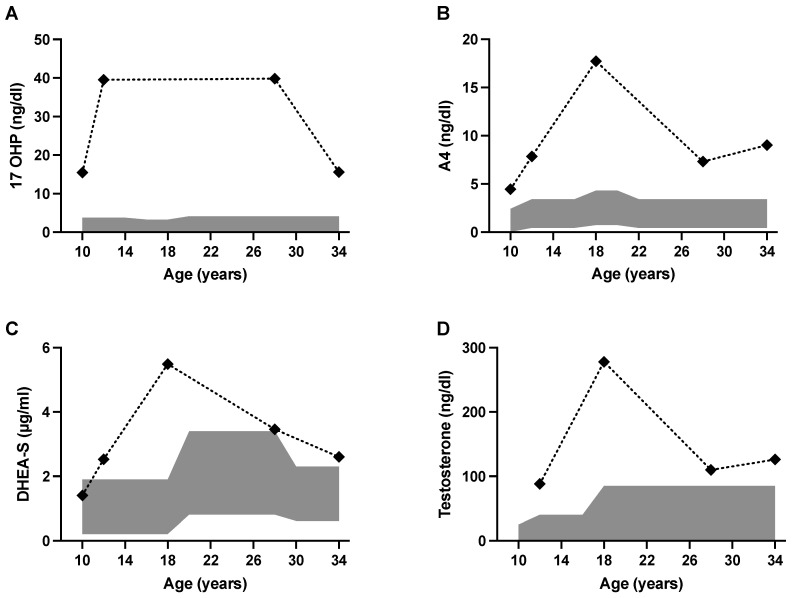
17-OHP and androgen levels of the patients at different ages.

**Figure 2 jcm-11-04307-f002:**
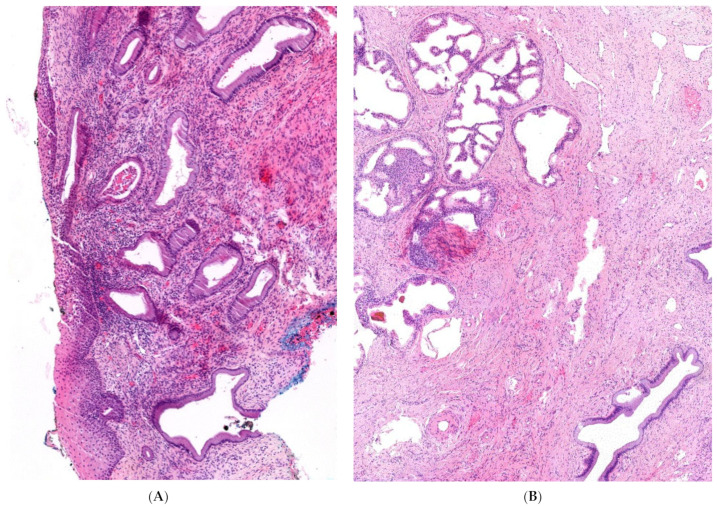
Histological features of the conisate. Histologic features of cervix tissue in HE staining with normal ecto- and endocervix in most specimens (**A**). Histologic features of ectopic prostate tissue with gland ducts in HE staining (**B**) and diffuse positivity in immunohistochemical testing with PSA (**C**) and NKX3.1 (**D**).

**Table 1 jcm-11-04307-t001:** Timeline of patient’s medical history.

Patient Visits	Patient Age	Clinical Presentation	Diagnostic Findings	Consequences and Procedure
Visit 1 10/1997	10	- Advanced bone age - Pubes: Tanner Stage V - Breast development: Tanner Stage V - Menarche at 10 - Mild acne - Normal external genitalia	- Premature development - Hyperandrogenaemia and elevated 17-OHP in serum - Genetic confirmation of non-classic CAH (compound heterozygous I2G/P30L)	- Starting hydrocortisone therapy 20 mg/day
Visit 2 05/2000	12	- Normal external genitalia - Mild acne - Regular cycle	- Hyperandrogenaemia and elevated 17-OHP level in serum - Elevated pregnanetriol in a 24-h urine sample	- Continuation of hydrocortisone therapy with 20 mg/day
Visit 3 07/2006	18	- No Acne - Mild clitoral hypertrophy	- Hyperandrogenaemia and elevated 17-OHP level in serum	- No therapy at patient’s request
Visit 4 05/2016	28	- Hirsutism - Regular cycle	- Hyperandrogenaemia and elevated 17-OHP level in serum	- Starting prednisolone therapy with 2 mg/day to achieve pregnancy
Visit 5 10/2021	34	- Hirsutism - Scarring acne	- Hyperandrogenaemia and elevated 17-OHP level in serum - Pathological cervical smear: (HPV)-positive dysplasia	- Scheduling a conization
Visit 6 01/2022 Post-conzation follow-up	34	- Hirsutism - Scarring acne	- Hyperandrogenaemia and elevated 17-OHP level in serum - Ectopic, metaplastic prostatic tissue in the cervix uteri (resected in total)	- Starting therapy with modified release hydrocortisone 10mg/day and an anti-androgenic oral contraceptive

Depicted is the patients’ medical history with clinical and diagnostic findings at each visit at our clinic, as well as treatment and further procedure. 17-OHP 17-Hydroxyprogesterone A4 Androstenedione DHEA-S Dehydroepiandrosterone sulfate.

## Data Availability

The data of this case are available from the corresponding author upon reasonable request.

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
