# Peer review of "Ectopic Prostate Tissue in the Uterine Cervix of a Female with Non-Classic Congenital Adrenal Hyperplasia—A Case Report"

_jcm, 2022, doi:10.3390/jcm11154307_

Round 1

Reviewer 1 Report

The manuscript by Tschaidse and co-workers is an interesting, well written and comprehensively worked-up case report on a female patient with genetically proven, poorly controlled non-classical CAH, in whom ectopic prostate tissue was detected following an abnormal cervical smear test. The report is novel in a sense as this has not been reported in the non-classic form of CAH, but is a known entity in classic forms of the condition. 

The data are well presented and the paper is very well written, appropriately in length and the discussion sparks interesting thoughts and considerations including as to how excess androgens in even more common conditions (i.e. PCOS) might contribute to cervical ectopic developmental prostatic remnants. It might be worth also mentioning in that context other, better known developmental remnants such as testicular adrenal rests in boys/men or ovarian adrenal rests in women. The latter, has this also been looked at in the reported case?

 Congratulate the team for this well written and informative report.    

Author Response

jcm-1809349

Ectopic prostate tissue in the uterine cervix of a female with non-classic congenital adrenal hyperplasia

  Dear editors, dear reviewers,

 We appreciate the careful reviewing of our case report and are grateful for the reviewers’ comments and suggestions which have helped to improve the case report.

We addressed the reviewers’ comments in the attached point-to-point reply and figure. We are convinced that we provide detailed responses to all concerns and hope that our case report will now be acceptable for publication in the Journal of clinical medicine.

We are looking forward to your feedback.

Yours sincerely,

Nicole Reisch on behalf of all co-authors

Reviewers’ comments to authors:

Reviewer 1

The manuscript by Tschaidse and co-workers is an interesting, well written and comprehensively worked-up case report on a female patient with genetically proven, poorly controlled non-classical CAH, in whom ectopic prostate tissue was detected following an abnormal cervical smear test. The report is novel in a sense as this has not been reported in the non-classic form of CAH, but is a known entity in classic forms of the condition.

The data are well presented and the paper is very well written, appropriately in length and the discussion sparks interesting thoughts and considerations including as to how excess androgens in even more common conditions (i.e. PCOS) might contribute to cervical ectopic developmental prostatic remnants. It might be worth also mentioning in that context other, better known developmental remnants such as testicular adrenal rests in boys/men or ovarian adrenal rests in women. The latter, has this also been looked at in the reported case?

Congratulate the team for this well written and informative report.   

Reply: We thank the reviewer very much for the appreciation of our case report. As suggested by the reviewer, we have added a sentence with regard to other developmental remnants such as testicular and ovarian adrenal rest tissue in the discussion (lines 195-198).

Reviewer 2

The work by Tschaidse et al. reports a case of ectopic prostate tissue in the uterine cervix of a female with NC congenital adrenal hyperplasia.

There are some things that need to be improved in order to publish the work.

  1. Please adjust the format of the table, it is too big. Also, it is not attractive to the eye, this table has not a lot of information and should be presented in a more attractive way.

Reply: We thank the reviewer for this comment. We have improved the way of presenting the data from the table by the following means: We replaced the original table with the laboratory values by a table (Table 1) with a treatment timeline instead (As required by CARE guidelines, which was requested by Reviewer 3). The important laboratory values (17-OHP, A4, DHEA-S, testosterone) are still shown as graphs in Figure 1, other important parameters are mentioned in the text.

  1. You should remove one of the measurements performed at 34 years old, there's no need to include both, specially because they are quite similar. It can only confuse the reader.

Reply: Thank you for this important comment, indeed it does not add extra knowledge. We have removed one of the measurements as suggested by the reviewer.

  1. In the table also, you could add a line with the treatment.

Reply: We have included the information with regard to treatment in the new Table 1 "Timeline of treatment".

  1. Regarding PSA measure, you state in the manuscript that 2 measures were performed, before and after intervention. However in the table there is only one.

Reply: As Table 1 has been removed, both PSA measurements are now stated in the manuscript (lines 129-130).

  1. The figures are misleading. On the x axis there is no scale, just the different time points (which are not constant). Please adjust. Also, for 17-OHP there was not measurement at 18 years old, you can't just put a line between 12 and 28, you can draw dots for example. Similarly, for testosterone there is no measurement at 10 years old... This should be addressed.

Reply: We thank the reviewer for this helpful comment. According to the suggestion, we have converted the X-axis of Figure 1 to a constant scale (age in years, where 1 space corresponds to 4 patient years). Also, we added rectangular symbols to highlight the individual measurements, while the connecting lines are now dotted as recommended by the reviewer.

  1. Finally, in the discussion section, you state that malignant neoplasms of prostate tissue in these patients are possible. Apart the work from Wasselious, has this been described in more patients?

Reply: We conducted another search of pubmed and we only found the already described case of Wesselius and Schotman on this subject. We have thus added some wording to emphasise this (line 142/143).

Reviewer 3

Dear Author

You need to have written informed consent from your case to agree with you to present her case, and report your case based on CARE guidelines as well. Some mild English errors should be solved as well.

Reply: Thank you for the important comment about the patient's written informed consent. We have of course obtained this and have now also indicated it in the case report (lines 75-78). We have also adapted the case report and revised it according to the CARE Guidelines and given this information in the manuscript (line 257). Lastly, we have performed a language check. We hope that we have successfully implemented all of the above.

Reviewer 2 Report

The work by Tschaidse et al. reports a case of ectopic prostate tissue in the uterine cervix of a female with NC congenital adrenal hyperplasia. 

There are some things that need to be improved in order to publish the work. 

1. Please adjust the format of the table, it is too big. Also, it is not attractive to the eye, this table has not a lot of information and should be presented in a more attractive way. 

2. You should remove one of the measurements performed at 34 years old, there's no need to include both, specially because they are quite similar. It can only confuse the reader.

3. In the table also, you could add a line with the treament.

4. Regarding PSA measure, you state in the manuscript that 2 measure were performed, before and after intervention. However in the table there is only one. 

5. The figures are missleading. On the x abscis there is no scale, just the different time points (which are not constant). Please adjust. Also, for 17-OHP there was not measurement at 18 years old, you can't just put a line between 12 and 28, you can draw dots for example.  Similarly, for testosteron there is no measurement at 10 years old... This should be addressed. 

6. Finally, in the discussion section, you state that malignant neoplasns of prostate tissue in these patient are possible. Apart the work from Wasselious, has this been described in more patients?

Author Response

(The authors gave the same response as above.)

Reviewer 3 Report

Dear Author

You need to have written informed consent from your case to agree with you to present her case, and report your case based on CARE guidelines as well. Some mild English errors should be solved as well.

Author Response

(The authors gave the same response as above.)
